# Wood-Based Cellulose-Rich Ultrafiltration Membranes: Alkaline Coagulation Bath Introduction and Investigation of Its Effect over Membranes’ Performance

**DOI:** 10.3390/membranes12060581

**Published:** 2022-05-31

**Authors:** Anastasiia Lopatina, Alma Liukkonen, Sabina Bec, Ikenna Anugwom, Joona Nieminen, Mika Mänttäri, Mari Kallioinen-Mänttäri

**Affiliations:** Department of Separation Science, LUT School of Engineering Science, LUT University, P.O. Box 20, FI-53851 Lappeenranta, Finland; alma.liukkonen@student.lut.fi (A.L.); sabina.bec@lut.fi (S.B.); ikenna.anugwom@lut.fi (I.A.); joona.nieminen@lut.fi (J.N.); mika.manttari@lut.fi (M.M.); mari.kallioinen-manttari@lut.fi (M.K.-M.)

**Keywords:** membrane fabrication, cellulose, wood, ultrafiltration, wastewater treatment, phosphorus removal, alkaline treatment, 1-ethyl-3-methylimidazolium acetate, choline chloride, lactic acid

## Abstract

In this study, wood-based cellulose-rich membranes were produced with a novel approach to casting procedure. Flat-sheet membranes were prepared from birch biomass pretreated with deep eutectic solvent and dissolved in ionic liquid-dimethylsulfoxide system via phase inversion method. Alkaline coagulation bath filled with sodium hydroxide solution was added to the process before a water coagulation bath and aimed to improve membranes’ performance. The effect of NaOH coagulation bath on the membrane was studied based on two NaOH concentrations and two different treatment times. The characterisation methods included measuring pure water permeabilities, polyethylene glycol 35 kDa model solution retentions, hydrophilicity, zeta potential, and chemical structure. Additionally, suitability of the membranes for removing residual phosphorous from a municipal wastewater treatment plant’s effluent was studied. The study revealed that introduction of the alkaline coagulation bath led to additional removal of lignin from membrane matrix and increase in the filtration capacity up to eight times. The resulting membranes can be characterised as very hydrophilic, with contact angle values 11.9–18.2°, negatively charged over a wide pH range. The membranes with the highest permeability, 380–450 L/m^2^·h·bar, showed approximately 70% phosphorus removal from purified wastewater, good removal of suspended solids, and low irreversible fouling tendency.

## 1. Introduction

The significance of cellulose-based pressure-driven membrane filtration has been extensively described in literature [1,2,3,4,5]. Despite the chosen method of describing cellulose as a polymer—regardless of whether it is recognised as hydrophilic or an amphiphilic one [5,6,7,8,9]—the regenerated cellulose membranes are universally acknowledged as very hydrophilic, having higher biocompatibility and lower fouling tendency when compared with petroleum-based polymers due to cellulose being a biopolymer [10,11,12,13]. The existing research focusing on the manufacturing of cellulose membranes is highly variable regarding whether research groups prefer to use commercially produced cellulose [14,15,16] or bio-based cellulose sources [2,17,18], using various solvents with high capacity for cellulose dissolution [3,4,19]. The steady interest towards ionic liquids (ILs) as cellulose and biomass solvent media is supported with such characteristics as their potential recyclability, low vapour pressure, and good chemical and thermal stability [20,21,22,23]. The challenges regarding ILs’ high price and high viscosity can be mitigated successfully by the addition of a co-solvent [13,16,21,24,25,26,27].

In a previous study [18] we showed that it is possible to produce membranes from carbohydrates originating from birch. However, the membrane retention and filtration capacity that were achieved in that process were not at the level required for a good, commercially feasible membrane. It is known that changing the fabrication procedure conditions enables controlling the membrane morphology and surface properties, which then significantly influences membrane performance [28]. Thus, the possibilities to improve the performance of the birch-based cellulose-rich membranes were studied from the related literature. The use of dilute acidic solutions as the coagulation bath media is well studied [29]; however, the use of alkaline solutions for the coagulation purposes seemed to be lacking attention. Cellulose behaviour in alkali systems is well studied for the case when alkaline solutions are used for the dissolution or extraction of cellulose [30,31]. However, this information can only be partly applied to the needs of membrane manufacturing, considering the backwards direction of the process. Thus, we chose to focus on the possibility to improve the performance of the birch-based cellulose-rich membrane via the use of an additional coagulation bath filled with alkaline solution.

Although there are countless combinations of alkali with additive systems (e.g., NaOH and urea, NaOH and PEG, KOH as a NaOH substitute, etc.), when pure NaOH is used for the cellulose dissolution, the range of applicable concentrations is very specific (7–10 wt.% below −5 °C) [6]. There are studies reporting the dissolution of cellulose in an NaOH solution of lower concentrations; however, the cellulose described there is usually amorphous and has a degree of polymerisation (DP) ≤200. It is also important to remember that the presence of lignin affects the dissolution process of cellulose in alkaline solutions, usually making it easier [9]. Lignin dissolution by itself requires a significantly lower NaOH concentration, starting from an approximately 1 wt.% solution [32,33,34]. Hemicelluloses usually require slightly higher NaOH concentrations (starting at 4 wt.%) but still lower ones than needed for the cellulose dissolution [35]. Thus, the concentrations of interest were picked in order to have: (1) a lower concentration (5 wt.%) of NaOH at which the dissolution of lignin and hemicelluloses should happen but at which cellulose is left intact; and (2) a higher concentration (10 wt.%) of NaOH at which there might be observable changes to cellulose structure [36,37].

The wood-based cellulose-rich membranes were prepared via a non-solvent induced phase inversion method from birch biomass purified from lignin with deep eutectic solvent (DES), consisting of choline chloride (ChCl) acting as a hydrogen bond acceptor (HBA) and lactic acid (LAc) acting as a hydrogen bond donor (HBD). The treated biomass was dissolved in a 1-ethyl-3-methylimidazolium (Emim) acetate (OAc) dimetylsulphoxide (DMSO) system; the detailed procedure can be found in our previous publication [18]. The casting procedure was improved with the addition of a second coagulation bath, filled with sodium hydroxide solution, that aimed to improve the performance of the membranes. The membranes’ permeability was measured in both dead-end and cross-flow membrane modules. Membranes’ retention properties were characterised with both the filtration of model solutions (polyethylene glycol (PEG), CaCO_3_) and the filtration of purified municipal wastewater to remove phosphorous. Additionally, the hydrophilicity, zeta potential, and chemical structure of the membranes were characterised in order to examine the influence of the alkaline coagulation bath.

## 2. Materials and Methods

### 2.1. Materials

Debarked birch (*Betula pendula*) woodchips with a nominal size of 5 × 1 × 0.1 cm were used as a base material for all further operations. Partial delignification of the birch chips was done via DES treatment, where choline chloride (CAS # 67-48-1, Merck KGaA, Darmstadt, Germany) was used as an HBA, and lactic acid (CAS # 79-33-4, Merck KGaA, Darmstadt, Germany) was used as an HBD. A mixture of ethanol and deionised (DI) water (15 MΩ, 0.5–1 μS/cm) at a 9:1 volume ratio was used for washing out the residual DES from the partially delignified biomass. A mixture of ionic liquid–1-ethyl-3-methylimidazolium acetate, 95% (C_1_C_2_ImOAc, CAS # 143314-17-4, Iolitec Ionic Liquids Technologies GmbH) and dimethylsulphoxide (CAS # 67-68-5, Merck KGaA, Darmstadt, Germany) was used for preparation of the casting solution. Sodium hydroxide pellets (>99%, CAS # 1310-73-2, Merck KGaA, Darmstadt, Germany) were used for membrane precipitation in the preparation of alkaline solutions.

Flat-sheet membranes were cast on nonwoven polypropylene/polyethylene carrier material (Viledon^®^ Novatexx 2484, Freudenberg, Germany; air permeability: 60 L/(s·m^2^); weight per unit area: 85 g/m^2^; maximum tensile force along/across: 300/200 N/5 cm; 25/30% elongation at maximum tensile force along/across; and thickness: 0.12 mm). Ultra-pure DI water (15 MΩ, 0.5–1 μS/cm) was produced by a CENTRA-R 60/120 system (Elga purification system, Veolia Water, London, UK) and used for washing, as a non-solvent, and in the preparation of all solutions.

Polyethylene glycol (approximately Mw 35,000 g/mol, CAS # 25322-68-3, Merck KGaA, Darmstadt, Germany) was used as a model compound for the study of membrane retention. Calcium carbonate (powder, ≤50 μm particle size, >98%, CAS # 471-34-1) was used as a model compound in the experiments showing the removal of suspended solids with the produced membranes. The purified wastewater used in the experiments was the discharge water of the Toikansuo wastewater treatment plant’s secondary clarifier, located in Lappeenranta, Finland. The Toikansuo plant’s treatment capacity is approximately 20,000 m^3^/day. The purified wastewater’s pH was 7.75, its total carbon concentration was 5.92 mg/L, its total nitrogen concentration was 9.4 mg/L, and its total phosphorous concentration was 0.35 mg/L. The performance of the birch-based membranes produced in this study was compared with the performance of the commercial membrane RC70PP (Alfa Laval; support material: polypropylene; membrane matrix: regenerated cellulose acetate; molecular weight cut-off (MWCO) 10,000 kDa).

### 2.2. Methods

#### 2.2.1. DES Treatment

For treatment of the birch woodchips, DES was prepared from ChCl and LAc mixed at a 1:9 mole ratio, respectively, at 100 °C until a homogeneous transparent mixture formed. The birch chips and DES were taken at a 1:5 mass ratio and placed into an oil bath, wherein the biomass load was cooked for 18 h at 105 °C. After the treatment was completed, the pulp was vacuum filtered through filter paper and washed with a mixture of ethanol and DI water at a 9:1 volume ratio until the treated pulp released no colour. The DES-treated pulp was transferred to an oven and dried at 50 °C for 24 h.

#### 2.2.2. Membrane Preparation

Homogeneous solutions of the DES-treated biomass samples were prepared in a mixture of Emim OAc and DMSO with 5 wt.% concentration by constant stirring overnight under constant heating in an oil bath at 90 °C. The prepared solutions were cooled down to room temperature and cast on the carrier material placed on a glass plate by spreading an appropriate amount with Automatic Film Applicator L (BYK-Gardner, Pompano Beach, FL, USA) with a 150 μm casting knife at a speed of 50 mm/s. Immediately after casting, the membrane was transferred into the first coagulation bath filled with NaOH solution and, after that, it was moved into the second coagulation bath filled with DI water at 0 °C, where the membrane was preserved for 24 h to guarantee a complete phase separation (see Figure 1).

In addition to the alkaline solutions strength justification discussed in the Introduction, it is vital to mention that cellulose solutions in cold alkali are recognised as metastable (i.e., sensitive to the slightest pH or temperature changes and aging, all of which can trigger the gelation-regeneration process) [6,8,24]. Naturally, when cellulose behaviour in alkali was studied, the conditions were maintained as stable as possible; however, when the current case of a coagulation bath is discussed, there is no need to stabilise the cellulose’s state in the alkaline solution, thus only the starting temperature of the alkaline solution was adjusted to 0 °C. During the precipitation process, the temperature was allowed to rise in a natural way. The obvious assumption here would be that cellulose precipitation should be described as delayed (both due to the low temperature and the alkaline conditions) but nonetheless surely existing in the alkaline solution [6,38].

Regarding the residence time in the coagulation bath, it was chosen based on the reported cellulose gelation rates. Although various researchers stated different gelation rates for the solutions of biomass, it is generally acknowledged that, within the range of concentrations and time intervals chosen for the experiments, the gelation is supposed to happen within 30–40 min, depending on the temperature [38]. Thus, two time intervals were tested: (1) a shorter one (30 min), the alleged minimum time required for gelation process; and (2) a longer one (60 min), when gelation surely happened. Taking into consideration that the temperature of the alkaline solutions was allowed to rise naturally, it can be safely assumed that cellulose precipitation should be finished by the time the coagulation bath is changed in both time cases. Table 1 summarises the casting parameters’ variations and gives codes for the membrane types produced.

All the obtained membranes were washed under DI water and then stored in water and used for analyses without drying if not stated otherwise. An overview of the analyses of the prepared membranes is presented in Table 2.

#### 2.2.3. Membrane Permeability and Retention Measurements

The permeability and retention of the prepared membranes were measured with Amicon dead-end stirring cell equipment (Millipore, Burlington, MA, USA, Cat No.: XFUF07611; diameter of the stirring device: 60 mm) (see Figure 2). Before the filtration experiments, each membrane was compacted for 1 min at 1 bar, 2 min at 2 bars, 3 min at 3 bars, 4 min at 4 bars, and 20 min at 5 bars. This also ensured that the solvents used in the membrane manufacturing were completely rinsed from the membrane pores. This was also checked through the measurement of total organic carbon (TOC) content in a permeate sample collected during the membrane compaction at 5 bars.

The pure water permeability of the membranes was determined as a slope of four plotted flux values, measured at 1, 2, 3, and 4 bars of pressure at 25 ± 0.5 °C and calculated using Equation (1):(1)$$J=\frac{Q_{P}/1000}{A\cdot \frac{\tau }{60}},$$
where ***J*** is the tested membrane’s flux (L/(m^2^·h)), ***Q_P_*** is the gravimetric flow of water permeating through the membrane (g/min), ***A*** is the area of the membrane sample (m^2^), and ***τ*** is the time of collection of the permeate (min).

Using the 10/30 and 10/60 membranes, the permeate flux of the purified wastewater was followed as a function of time. The total filtration experiment took place for 11 days with a volume reduction factor (VRF; the volume ratio between the permeate and the initial feed) of about 3. Additionally, the pure water permeabilities were determined before and after the long wastewater filtration. The filtration experiments with wastewater were performed in a cross-flow filtration module, with the size of a single membrane coupon of 10.4 cm^2^ (see Figure 3). The cross-flow velocity used in the wastewater experiments was *v* = 1.2 m/s, measured at 1 bar pressure and 20 ± 0.5 °C.

For studying the retention of the produced membranes, a model solution of PEG 35 was prepared with a concentration of 300 ppm and filtered through the membrane at a pressure that was set for each membrane individually in order to have approximately the same flux. Throughout all the measurements, the stirring speed was maintained at 300 rpm using the magnetic stirrer with an rpm indicator. The temperature was kept at 25 ± 0.5 °C. The samples of the feed, retentate, and permeate were collected and analysed for total organic carbon (TOC) content with a Shimadzu TOC analyser (TOC-L series, Kyoto, Japan). Membrane retentions were calculated out of the measured TOC content in the samples using Equation (2):(2)$$R=\left(1-\frac{2\cdot C_{p}}{C_{f}+C_{r}}\right)\cdot 100,$$
where ***C_p_***, ***C_f_***, and ***C_r_*** are the TOC concentrations in the permeate, feed, and retentate (mg/L), respectively.

To measure the removal of suspended solids, filtration experiments with CaCO_3_ (1000 ppm) solution were conducted. The turbidity of the samples was measured with the Hach Model DR/2010 Spectrophotometer by means of the preinstalled programme #750. The removal efficiency was calculated using Equation (2).

The purified wastewater from the Toikansuo wastewater treatment plant’s secondary clarifier was used for the assessment of total phosphorus removal with the manufactured membranes. The samples were analysed with the Spectroquant^®^ tubes kit (o-phosphate and total phosphorous), using the photometric method: PMB 0.05–5.00 mg/L, PO_4_− P 0.2–15.3 mg/L, PO_4_^3^− 0.11–11.46 mg/L P_2_O_5_. The photometer used was a NOVA 60 A Spectroquant^®^. The removal efficiency was calculated using Equation (2).

#### 2.2.4. Membrane Characterisation

##### Examination of the Zeta Potential of the Membranes

The zeta potential was measured with a SurPASS electrokinetic analyser (Anton Paar GmbH, Graz, Austria) with an adjustable gap cell method and using 0.001 M KCl solution as a background electrolyte. The membranes were preliminary stored at approximately 5 °C for 24 h in a fridge. Before the start of the experiment, the electrolyte solution’s pH was shifted to 7.6 by the addition of 0.1 M KOH solution and then automatically titrated to 2.4 with use of a 0.05 M HCl solution as the analysis carried on. The final value of the zeta potential was calculated automatically by SurPASS software, based on the Helmholtz–Smoluchowski equation. In addition to zeta potential values, the isoelectric point (IEP), the pH value where the zeta potential equals 0 mV, was identified.

##### Examination of the Chemical Structure of the Membranes

The Fourier-transform infrared spectroscopy (FTIR) spectra of membrane samples were measured with a Frontier MIR/FIR spectrometer (PerkinElmer Inc., Waltham, MA, USA) with the universal diamond crystal attenuated total reflectance (ATR) module in the range of wave numbers 400–4000 cm^−1^, with the spectra resolution of 4 cm^−1^. Air-dried samples of the selected membranes were recorded in pentaplicate and averaged. For the graphical representation, all the spectra were processed with ATR correction, baseline correction, and normalisation. The ratio of the not-normalised absorption bands A_1428_/A_899_ was used to calculate a lateral order index (LOI), as was proposed by Nelson and O’Connor [39,40].

##### Examination of the Hydrophilicity of the Membranes

The membranes’ hydrophilicity was measured with a static contact angle (CA) procedure based on the captive bubble method [41]. The U-shaped needle placed nearly 3–4 μL of air bubble volume on the surface of the tested membrane, attached to a piece of glass with double-sided tape and submerged into DI water at room temperature. For each membrane sample, six independent measurements of the CA were made at different points with the average value of recorded data taken as the final CA. The CA was measured with KSV CAM 101 equipment (KSV Instruments Ltd., Espoo, Finland) connected to a charged-coupled device (CCD) camera (DMK 21F04, Imaging Source Europe GmbH, Bremen, Germany). To determine the CA, the obtained images were treated by curve-fitting analysis with CAM 2008 software.

## 3. Results and Discussion

### 3.1. Changes in Membrane Characteristics Due to the Alkaline Treatment

When the alkaline bath was introduced into the membrane manufacturing process, a common observation with all the condition and concentration combinations used in the experiments was the change of the first (alkaline) coagulation bath’s colour from transparent to brownish. This is a clear indication of partial lignin washing out from the membrane matrix into the coagulation bath [32]. Based on the FTIR spectra measured in this study, the changes happening to the cellulose surface, or biomass surface for that matter, in the used conditions were not significant (see Figure 4), which was reported earlier for similar treatment conditions [36].

Comparing the reference spectrum (0/0) with the membranes prepared with the alkaline bath in the production line, small changes in the 900–1100 cm^−1^ region and consistent changes of the wave numbers in the 1200–1700 cm^−1^ region can be seen. The presence of the peak at 1710 cm^−1^ in all the spectra is indicative of the introduction of C=O groups as a result of the cellulose’s partial acetylation during the dissolution process in IL, which has been previously reported and attributed to either overall biomass complexity or the presence of water residuals [37,42]. Comparison of the 0/0 membrane with the other membranes that started the coagulation process in the alkaline bath shows an increase of the peak at 899 cm^−1^ due to alkaline treatment, which is a characteristic peak of C–O stretching vibration in the amorphous region of cellulose, implying an increase in the total amount of amorphous cellulose in the membrane matrix [43,44]. The easiest way to reaffirm crystallinity assessment is the calculation of either the crystallinity index or the LOI, as was suggested by Nelson and O’Connor [39,40]. The LOI is revealed from the ratio at 1428 cm^−1^ in-plane symmetric bending, characteristic for cellulose I_β_ crystal, and C–H deformation in β-glycosidic linkages at 899 cm^−1^, specific for amorphous cellulose regions (A_1428_/A_899_) [45,46]. The calculation showed the distinct difference between membranes coagulated using the alkaline coagulation bath and the 0/0 membrane: the membrane coagulated in water shows a LOI value of 0.82 whereas all the other membranes show a LOI value within 0.62–0.65, definitely showing a smaller content of crystalline cellulose and thus a larger share of amorphous region. According to results previously reported by other researchers, the decrease in the crystallinity of the membrane would lead to a decrease in the membranes’ tensile properties and a simultaneous increase in the elongation break values [12,47]. These changes might have an impact on the usability of a membrane; however, no noticeable difference was observed in the scope of this study. The small decrease, which can be observed at wave numbers 1240 and 1507 cm^−1^, can be assigned to the different units of lignin, which suggests additional removal of lignin in the alkaline coagulation bath [32,48,49,50].

The CA measurements show that all the membranes can be characterised as hydrophilic (0/0) membranes or super-hydrophilic membranes (5/30, 5/60, 10/30, 10/60 membranes) (see Figure 5). This was expected due to the fact that hydrophilicity is common for cellulose membranes as regenerated cellulose films are known to be amongst the most hydrophilic polymers [38]. Generally, the CAs of alkali-treated membranes are noticeably smaller than those coagulated in DI water, which correlates well with FTIR measurements, suggesting that similar changes are happening within the membrane matrix from the chemical point of view (i.e., there is additional removal of lignin within the matrix). The zeta potential results demonstrate that all the membranes are negatively charged over a wide range of pH (see Figure 5). The membranes demonstrate stronger negative zeta potential at a neutral pH as a stronger NaOH concentration and longer time were applied in the first (alkaline) coagulation bath.

The more negative zeta potential values might be the result of the removal of non-cellulosic compounds, which makes the cellulose’s surface groups more accessible for detection. It can also be speculated that the charge and hydrophilicity become stronger as the cellulose crystallinity reduces, leading to the increased amount of voids and flexible parts within the cellulose fibres and, thus, the better accessibility of the surface groups [36].

### 3.2. Membrane Performance

The results revealed that introducing the alkaline bath in the membrane manufacturing process improves the filtration capacity of the prepared cellulose membranes (see Figure 6a). The filtration capacities of the prepared membranes were actually increasing so much that the capability of the membranes to retain suspended solids was tested with a CaCO_3_ suspension. It was found that the 10/60 membranes retained more than 97% of the suspended CaCO_3_ particles. Although the filtration capacity change was significant, the retention of PEG molecules was at the same level for the reference membrane (0/0), for the membranes prepared with the 30-min alkaline bath treatment (5 and 10%), and for the membrane prepared with the longer alkaline treatment in the 5% alkaline bath (see Figure 6b).

As the permeability of ultrafiltration membranes prepared from semicrystalline polymers is highly dependent on the ratio between crystalline and amorphous parts, it is generally acknowledged that higher amorphous content results in better permeability [51]. This correlates well with the zeta potential measurements, where the strength of the charge on the membrane surface at neutral pH was directly proportional to the strength of the alkaline bath used.

Based on the high permeabilities, and thus the potential to be scaled up, the 10/30 and 10/60 membranes were taken onto further tests with the cross-flow module and their performance is compared with the commercially available RC70PP membrane (Alfa Laval). The results showed that the removal efficiency for the residual phosphorous from the effluent of the municipal wastewater treatment plant were at the same level with all membranes (see Table 3). The sufficient removal of phosphorus from anthropogenic wastewaters should be met by any treatment plant effluent before being discharged to the surface water. According to the present regulations, the maximum allowable amount of phosphorus in the purified wastewater effluent in Lappeenranta is 0.5 mg P/L [52,53]. However, future regulations will limit the total phosphorus concentration to below 0.1 mg P/L [54]. The effluent used in the filtrations contained phosphorous levels of about 0.25 mg/L. The phosphorous concentration in the permeate of the studied membranes was around 0.09 mg/L and, thus, below the future regulation. Current removal techniques demand a large footprint and the use of additional chemicals to achieve appropriate removal efficiency. Based on the results shown here, one solution could be the introduction of the membrane filtration step at the end of the treatment line.

The filtration was made at a constant 1 bar pressure, which means that the measured initial fluxes of the 10/30 and 10/60 membranes were about two times higher (pure water fluxes: about three times higher) compared with the flux of the RC70PP membrane. This also means that concentration polarisation was stronger with the 10/30 and 10/60 membranes, leading to a significant decrease in flux. However, the permeate fluxes were still a bit higher with the 10/30 and 10/60 membranes at the end of the filtration than with the reference RC70PP membrane. The measurements of pure water fluxes before and after the filtration of the effluent showed that the pure water fluxes decreased more with laboratory-made membranes when compared with the RC70PP membrane. In addition, it was observed that the fluxes of the 10/30 and 10/60 membranes increased during water flux measurement.

A similar phenomenon was observed when the same membranes were used for wastewater filtration in the Amicon dead-end filtration module. The wastewater permeate flux declined by approximately 45% and 38% for the 10/30 and 10/60 membranes, respectively, when compared with the initial pure water permeability (PWP) value of those membranes. When the same membrane samples were simply rinsed with clean water, the permeability was restored to up to 60% and 95% of the initial PWP for the 10/30 and 10/60 membranes, respectively. This means that irreversible fouling of the membranes was low. Partial restoration of permeability is a known phenomenon of cellulose behaviour and sometimes linked to a negatively charged surface [55,56]. Since such membranes show the ability for partial permeability restoration, it means that not only the membrane has some intrinsic stability against fouling but also that cleaning procedures potentially do not require harsh chemicals, and it is possible to just clean them with water, thus reducing stress of the environment and the membrane over its life cycle.

The 10/30 and 10/60 membranes also showed similar removal of total phosphorous, 72% and 75%, respectively. The reason behind such good phosphorus removal with such open membranes might be that the residual phosphorus in the purified wastewater is due to the use of precipitation chemicals in the water treatment process in the form of very fine precipitates.

## 4. Conclusions

The fabrication of wood-based cellulose-rich membranes from DES-treated biomass solutions in a mixture of Emim OAc and DMSO, and using the phase inversion method was performed using a procedure involving two subsequent coagulation baths. The introduction of the additional coagulation bath filled with NaOH solution was done to improve the membranes’ performance. Two NaOH concentrations (5 and 10 wt.%) and two residence times (30 and 60 min) were studied. Characterisation of the prepared membranes was done by measuring pure water permeability in both dead-end and cross-flow modules, as well as by measuring PEG 35 kDa retention, CaCO_3_ suspension separation, and the removal of total phosphorus from purified wastewater from a local treatment plant. Additionally, ATR-FTIR, static CA, and zeta potential measurements were carried out in order to study any changes occurring.

According to the results, the introduction of the alkaline coagulation bath into the casting procedure promotes the additional removal of lignin in the coagulation bath, while cellulose already starts the precipitation process. All the membranes prepared with the use of the NaOH bath showed higher hydrophilicity compared with the reference membrane (only precipitated in water) and a tendency to have a more negatively charged surface as stronger alkaline treatment was applied. The utilisation of the additional coagulation bath filled with NaOH solution proved to have a positive effect on the performance of the cellulose-rich wood-based membranes: they showed a significant increase of their filtration capacity alongside almost unchanged retention characteristics. Based on the CaCO_3_ retention (97%) and total phosphorus removal from purified wastewater (approximately 70%), these membranes might be an attractive alternative for use in the removal of residual phosphorous as a tertiary treatment at wastewater treatment plants due to the possibility to achieve a relatively good removal efficiency with a low pressure difference.

## Figures and Tables

**Figure 1 membranes-12-00581-f001:**
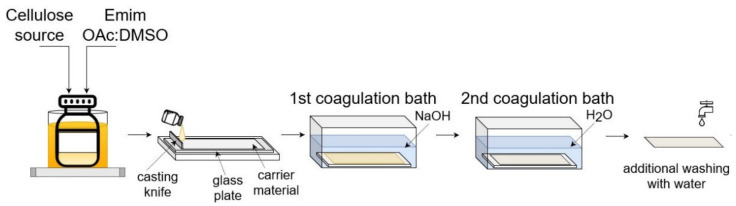
A schematic of the casting procedure.

**Figure 2 membranes-12-00581-f002:**
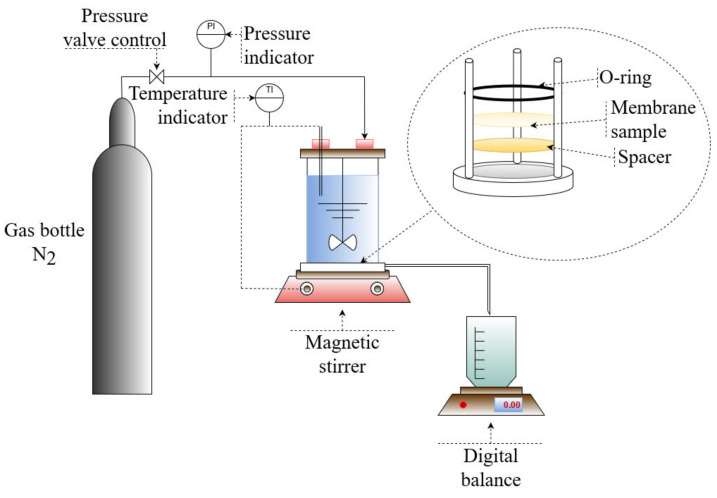
Schematic configuration of Amicon dead-end filtration system.

**Figure 3 membranes-12-00581-f003:**
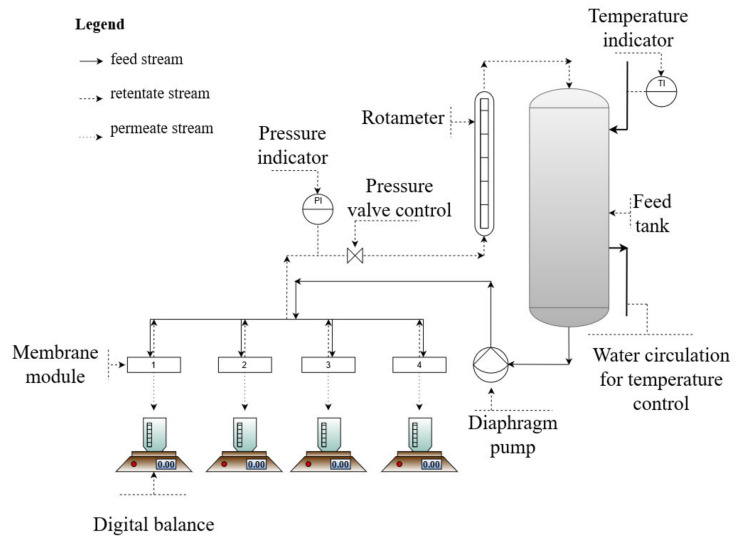
A schematic of the configuration of the cross-flow filtration system.

**Figure 4 membranes-12-00581-f004:**
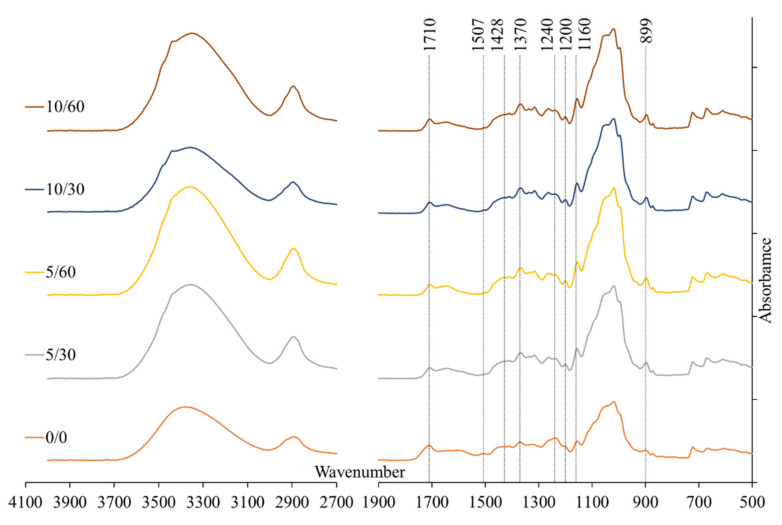
The FTIR spectra of the prepared membranes, recorded using the Perkin Elmer Frontier spectrometer with a universal ATR module of diamond crystal at a resolution of 4 cm^−1^ in the *absorbance* mode.

**Figure 5 membranes-12-00581-f005:**
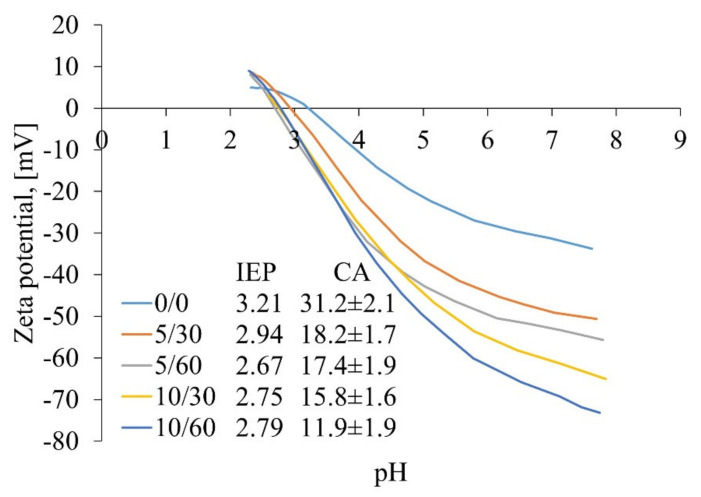
The zeta potential curves and isoelectric points (IEPs) of prepared membranes, recorded using a SurPASS electrokinetic analyser with the adjustable gap cell method and using 0.001 M KCl solution as an electrolyte; the contact angle values of the prepared membranes were recorded using the captive bubble method with KSV CAM 101 equipment connected to a CCD camera.

**Figure 6 membranes-12-00581-f006:**
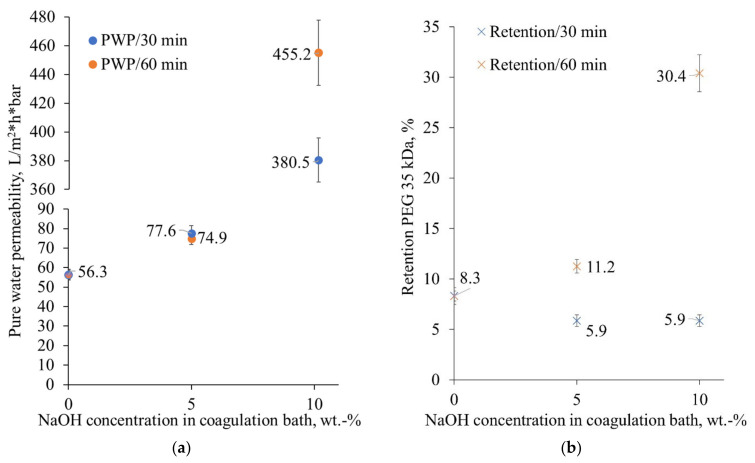
The pure water permeabilities (PWP) (**a**) and PEG 35 kDa retentions (**b**) of the tested membranes, all measured in the Amicon ultrafiltration cell at 25 °C and with a mixing rate of approximately 300 rpm.

**Table 1 membranes-12-00581-t001:** The casting parameters’ combinations and assigned membrane codes (0/0 for the membrane only coagulated in water).

NaOH Concentration, wt.%	Time Spent in NaOH Coagulation Bath, Min		
	**0**	**30**	**60**
0	0/0	-	-
5	-	5/30	5/60
10	-	10/30	10/60

**Table 2 membranes-12-00581-t002:** An overview of the analyses.

	FTIR	Zeta Potential	Amicon (Dead-End)				Cross-Flow		Contact Angle	P Tubes
			PWP	PEG Retention	CaCO_3_	WW Flux	PWP	WW Flux		
0/0	✔	✔	✔	✔					✔	
5/30	✔	✔	✔	✔					✔	
5/60	✔	✔	✔	✔					✔	
10/30	✔	✔	✔	✔		✔	✔	✔	✔	✔
10/60	✔	✔	✔	✔	✔	✔	✔	✔	✔	✔

**Table 3 membranes-12-00581-t003:** Pure water fluxes before and after effluent filtration, and permeate fluxes at the beginning and the end of the concentration filtration.

	10/60	10/30	RC70PP
Pure water flux before wastewater filtration at 2 bar, L/(m^2^h)	460	440	150
Wastewater flux at the beginning (8 min), L/(m^2^h)	106	105	64
Wastewater flux at VRF ~3 (11 days), L/(m^2^h)	69	68	42
Phosphorous removal, %	67	69	68

## Data Availability

The data presented in this study are available upon request from the corresponding author.

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
