# Peer review of "Wood-Based Cellulose-Rich Ultrafiltration Membranes: Alkaline Coagulation Bath Introduction and Investigation of Its Effect over Membranes’ Performance"

_membranes, 2022, doi:10.3390/membranes12060581_

Round 1
Reviewer 1 Report
The manuscript of Anastasiia Lopatina et al is devoted to the preparation of wood cellulose-based membranes from solutes in ionic liquids (IL). The manuscript has a classic look and is executed to a good standard. The authors consider the change in the transport properties of membranes depending on the method of obtaining the membrane. For the resulting membranes, structural features are evaluated using IR spectroscopy. For membranes, pure water permeability, retention of PEG (35 kDa), separation of CaCO3, and phosphorus removal are assessed.
When reading the manuscript, questions immediately arise: what is the degree of solvent regeneration (with a co-solvent) when using alkaline baths? As is known, ILs are toxic solvents and their residual content in the membrane is unacceptable, how was the degree of solvent removal from the membrane assessed?
In the list of references, I did not find the works of Prof. H. Sixta (https://scholar.google.es/citations?user=7JGBDowAAAAJ&hl=en), who has been dealing with IL and cellulose in Finland for many years. I recommend that the authors pay attention to this. The authors note the high cost of the solvent (lines 44-46), why then was the choice made in favor of it, and not, for example, N-methylmorpholine-N-oxide (NMMO)? I recommend the authors to compare these solvents and cellulose membranes obtained with their use.
Line 77. "wet phase inversion" - I recommend to rephrase.
Figure 1. It is necessary to increase the size of the inscriptions in the figure.
Why the authors call the 2nd coagulation bath. The process is not completed completely in the first bath? How was it evaluated? It is possible that the coagulation process has already been completed and the system is being changed by varying the washing baths (example - https://doi.org/10.1007/s10692-017-9786-x)
Table 1. What does 0/0 mean?
An increase in the amorphous fraction in cellulose, with a change in the production method, should undoubtedly lead to a decrease in the mechanical characteristics of the membrane. It is necessary to specify such data in the work.
Reviewer 2 Report
In this work, wood-based cellulose-rich membranes were produced with the novel approach to casting procedure. The preparation process is complex and the practical application performance is general. However, there are still many problems in this work, and further improvement is needed to meet the requirements of publication.
- One or two more sentences are suggested in the beginning of abstract to give the background of this work.
- The sentence “whether it is recognized as hydrophilic or an amphiphilic one” should be supported by an important and recent paper: whether it is recognized as hydrophilic or an amphiphilic one.
- Authors listed too many keywords, where no more than 6 keywords are required.
- The content in figure 1,2 is not clear. Adjust font size and improve image resolution. The quality and clarity of all the figures are poor. All the figures should be modified to provide a better resolution, and the word size of the picture should be adjusted.
- When describing the ionic liquids as solvent media, please carefully read and cite this important paper: Processing and valorization of cellulose, lignin and lignocellulose using ionic liquids.
- There have something wrong with the scale of Figures 4-6. There are no corresponding scale bars on the coordinate axes.
- (1, 2): there is no need to use italic font style for p in Qp, p in Cp or f in Cf, since they are not variables.
- How about the phosphorus removal mechanism?
- Can the adsorbed pollutant be recycled or used as a functional material for other applications?
- When general introducing the advantages of cellulose, one important and recent review paper should be carefully read and cited: A review on raw materials, commercial production and properties of lyocell fiber
- Multiple Casting procedure improvements and performance improvements are emphasized in this paper. But there is no corresponding comparison with other materials.
- When discussing the FT-IR results, necessary citations for the indications of peaks should be provided: Carbohydrate Polymers, Volume 276, 15 January 2022, 118799.
- Why author use wood derived cellulose should be clarified in the manuscript? In addition, can Debarked birch wood be replaced with other wood? Can other wood cellulose achieve the same effect?
Reviewer 3 Report
Comments and Suggestions for Authors
This paper focuses on developing a wood based cellulose flat sheet membranes via phase inversion method with an additional step comprising of alkaline coagulation with sodium hydroxide before water coagulation. The paper highlights the advantages of incorporating alkaline coagulation step to the standard water coagulation method by comparing membrane properties like permeability, hydrophilicity (contact angle measurements), zeta potential and chemical structure of the control membrane with the modified ones.
Overall, the paper is well-written and easy to follow. The research topic is relevant today and extensive research has been carried out by the authors to bring together this paper. I feel it is worthy of publication in the Membranes journal.
I have listed down a few comments which the author may consider for incorporating into the paper or commenting to make it even more comprehensive:
- Include more recent references: In the introduction, methods and discussion section include more recent and relevant papers from last 5-7 years to support your arguments and reasonings throughout the paper.
- Incorporate Scanning Electron Microscopy (SEM) images: The authors should incorporate SEM images of the blank or control membrane (at standard conditions) and compare its surface and cross section morphology with the membranes synthesized with NaOH coagulation at different conditions. Also, highlight and discuss any changes seen on incorporating NaOH coagulation step and mention the pore size and thickness for each membrane.
- Membrane reusability: With regards to membrane reusability, how many times can the synthesized membrane be reused without sacrificing on the removal performance? Can the authors comment on this?
- Mechanical strength testing of the membrane: If the authors want, they can incorporate a small section highlighting the tensile strength of the synthesized membranes.
- Pg 8, figure 5: Expand the acronym CA (Contact angle) and IEP (isoelectric point) or any other acronym in the main text of the paper before using it in the figures or tables.
Round 2
Reviewer 1 Report
The authors partially answered the questions posed by the reviewers. Unfortunately, information about the formed morphology and mechanical characteristics remained undisclosed. In my opinion, if the authors plan to publish this information in the next article, this is not a reason not to provide it for a response to the reviewers.
Below are additional recommendations:
keywords. I recommend removing or replacing "wood".
Authors need to fix links in places where a-umlaut is used, for example, Line 469.
Author Response
Response to comments of Reviewer 1
The authors partially answered the questions posed by the reviewers. Unfortunately, information about the formed morphology and mechanical characteristics remained undisclosed. In my opinion, if the authors plan to publish this information in the next article, this is not a reason not to provide it for a response to the reviewers.
Response: we did not mean gatekeeping of the information, the experiments on the morphology and mechanical characteristics are truly planned, however, they have not yet been performed, hence, we are not ready to share the information that we have not obtained yet.
Below are additional recommendations:
keywords. I recommend removing or replacing "wood".
Response: it would be interesting to know the reasoning behind this suggestion, however, due to the fact that wood remains a source of polymers for membrane matrix, and thus a significant part of novelty, the authors would like to keep it as a keyword.
Authors need to fix links in places where a-umlaut is used, for example, Line 469.
Response: with sincere respect, this comment is confusing for the authors. “Ä” (as well as “ö”) is an independent letter in the Finnish alphabet, which cannot be replaced, it is only simplified to “a” in very specific cases, e.g. in the email addresses. As for the links, the corresponding author has checked each link individually, and all of them direct the reader to the correct articles.
Reviewer 2 Report
Authors have addressed all the issues well. An acceptance is suggested.
Author Response
Response to comments of Reviewer 2
Authors have addressed all the issues well. An acceptance is suggested.
Response: all authors are grateful for the suggestions and acceptance.
Reviewer 3 Report
I am satisfied with the comments and arguments put forth by the authors and I believe the paper can now be accepted.
Author Response
Response to comments of Reviewer 3
I am satisfied with the comments and arguments put forth by the authors and I believe the paper can now be accepted.
Response: all authors are grateful for the suggestions and acceptance.